# New Insights into the Neurodegeneration Mechanisms Underlying Riboflavin Transporter Deficiency (RTD): Involvement of Energy Dysmetabolism and Cytoskeletal Derangement

**DOI:** 10.3390/biomedicines10061329

**Published:** 2022-06-06

**Authors:** Fiorella Colasuonno, Chiara Marioli, Marco Tartaglia, Enrico Bertini, Claudia Compagnucci, Sandra Moreno

**Affiliations:** 1Genetics and Rare Diseases Research Division, Bambino Gesù Children’s Hospital, IRCCS, 00165 Rome, Italy; fiorella.colasuonno@opbg.net (F.C.); chiara.marioli@opbg.net (C.M.); marco.tartaglia@opbg.net (M.T.); enricosilvio.bertini@opbg.net (E.B.); 2Department of Science, LIME, University Roma Tre, 00165 Rome, Italy

**Keywords:** riboflavin transporter deficiency, riboflavin, fatty acid oxidation, energy metabolism, oxidative stress, cytoskeleton, induced pluripotent stem cells, motor neuron disease, peroxisome, mitochondria

## Abstract

Riboflavin transporter deficiency (RTD) is a rare genetic disorder characterized by motor, sensory and cranial neuropathy. This childhood-onset neurodegenerative disease is caused by biallelic pathogenic variants in either *SLC52A2* or *SLC52A3* genes, resulting in insufficient supply of riboflavin (vitamin B2) and consequent impairment of flavoprotein-dependent metabolic pathways. Current therapy, empirically based high-dose riboflavin supplementation, ameliorates the progression of the disease, even though response to treatment is variable and partial. Recent studies have highlighted concurrent pathogenic contribution of cellular energy dysmetabolism and cytoskeletal derangement. In this context, patient specific RTD models, based on induced pluripotent stem cell (iPSC) technology, have provided evidence of redox imbalance, involving mitochondrial and peroxisomal dysfunction. Such oxidative stress condition likely causes cytoskeletal perturbation, associated with impaired differentiation of RTD motor neurons. In this review, we discuss the most recent findings obtained using different RTD models. Relevantly, the integration of data from innovative iPSC-derived in vitro models and invertebrate in vivo models may provide essential information on RTD pathophysiology. Such novel insights are expected to suggest custom therapeutic strategies, especially for those patients unresponsive to high-dose riboflavin treatments.

## 1. Introduction

Flavoproteome alterations have recently been associated with several rare diseases [1]. Among these, riboflavin transporter deficiency (RTD) is a severe childhood-onset neurodegenerative disorder, caused by inactivating mutations in genes encoding riboflavin (RF) transporters. In fact, RF, or vitamin B2, is an essential water-soluble vitamin and the precursor of flavin mononucleotide (FMN) and flavin adenine dinucleotide (FAD), which are coenzymes involved in a wide number of different metabolic processes occurring in mitochondria and peroxisomes, e.g., fatty acyl β-oxidation [2]. Fatty acids are a major fuel source used to sustain contractile function in the skeletal muscle; that the uptake and β-oxidation of fatty acids must be coordinately and finely regulated in order to meet its energy demands [3].

Following insufficient RF levels, changes occur in the cellular distribution of the various flavin fractions as well as in the activities of flavin-dependent enzymes. These changes suggest a specific hierarchic response to RF deficiency, e.g., the core electron transport chain (ETC) required for ATP synthesis is preserved, whereas the enzymes required for the first step of fatty acyl β-oxidation are reduced in their functionality [2].

Notably, a number of flavoenzymes are crucial for the mitochondrial oxidative phosphorylation (OXPHOS) function, including electron-transferring flavoprotein and electron-transferring flavoprotein-dehydrogenase, which transfer electrons from various reduced flavin groups to complex III via coenzyme Q10, and constituent subunits of Complexes I and II.

Similar to mitochondria, peroxisomes are also dynamic organelles that continuously adapt their number, morphology, and function to prevailing environmental conditions. In mammals, these organelles play a central role in many metabolic processes, some of which depend on flavoproteins. As such, it is not surprising that mitochondrial and peroxisomal dysfunctions have been proposed to contribute to the pathogenesis of numerous metabolic and neurodegenerative disorders, including RTD [4,5,6].

Both organelles are also major sources of reactive oxygen species (ROS), whose overproduction contributes to neuronal death [7,8,9]. Mitochondrial dysfunction and concurrent impairment in their turnover could therefore account for the specific vulnerability of motor neurons, which largely rely on oxidative metabolism for maintaining their large soma and intense neurotransmission activity. Notably, such pathomechanism in RTD is shared with many other degenerative conditions, affecting motor neurons, including amyotrophic lateral sclerosis (ALS) [10,11]. Mitochondrial- and peroxisomal-related oxidative damage may even contribute to cytoskeletal abnormalities. Indeed, our recent studies have emphasized the pathogenic role played by the microtubule’s derangement, caused by ROS overproduction not properly counterbalanced by antioxidant response. Additionally, mutated forms of RF transporters may impair their physical interaction with tubulins, thus contributing to their disorganization [6,12].

Overall, knowledge on pathophysiological mechanisms underlying rare genetic disorders is often hindered by lack of reliable and informative disease models, suited to experimental manipulation. These obstacles have partially been overcome by the induced pluripotent stem cell (iPSC) technology, so that several neurodegenerative, hematopoietic, metabolic, and cardiovascular disorders can now be studied through differentiation of disease-relevant cells from reprogrammed patient-specific cells [13].

## 2. Riboflavin Transporter Deficiency: From Genetics to Clinical Presentation

Starting from 2010, RTD, formerly known as Brown-Vialetto-Van Laere (BVVL) syndrome, was found to be caused by biallelic pathogenic variants in either *SLC52A2* or *SLC52A3* genes [14], and over 75 (mostly nonsense and missense mutations) in both genes have been associated with RTD. *SLC52A1*, *SLC52A2* and *SLC52A3* genes, highly conserved in humans, encode RF transporter proteins RFVT-1, RFVT-2 and RFVT-3, respectively. In a recent review, Jin and Yonezawa list and describe pathogenic variants of RFVT-1/*SLC52A1*, RFVT-2/*SLC52A2* and RFVT-3/*SLC52A3*. These also include intron variant, splice site, frameshift and synonymous [15]. Even dominantly acting *SLC52A1* variants and gene deletions have recently been reported in clinical cases and have causally been linked to placental transport defects [15,16]. Specifically, *SLC52A1* haploinsufficiency was identified in two asymptomatic women as the cause of biochemical abnormalities in their newborns resembling those of multiple acyl CoA dehydrogenase deficiency (MADD), a typical muscular lipid-storage disorder characterized by defects in the oxidation of fatty acids and amino acids leading to a clinically heterogeneous disease associated with metabolic acidosis, cardiomyopathy, liver disease, episodes of metabolic decompensation, weakness muscle, and respiratory failure [15,16].

RFVTs are predicted to have 11 transmembrane domains [17]. However, it is possible that RFVTs might exist in different isoforms due to alternative processing of each transcript [16]. These proteins are responsible for transporting RF across cell membranes and are widely distributed in tissues throughout the body, with RFVT-2 and -3 being mostly expressed in the intestines, brain, and spinal cord [11,18] or produced by the intestinal microbiota, to a lesser extent. Excess vitamin is excreted in the urine as RF or in form of other metabolites [1]. Circulating RF is bound both to plasma albumin and a subfraction of immunoglobulins with average plasma concentrations of 10.5, 6.6 and 74 nmol/L for RF, FMN and FAD, respectively [19] (Figure 1).

RTD is a debilitating, life-shortening, neurodegenerative genetic disorder characterized by motor, sensory and cranial neuropathy. In approximately 50% of affected individuals, RTD onset is observed early during childhood, and 95% of patients develop symptoms before adulthood. Common symptoms of RTD may include hearing loss, muscle and axial weakness, respiratory compromise (including diaphragm paralysis and sleep apnea), and pontobulbar palsy. Vision loss and sensory gait ataxia are also commonly noted in RTD patients (Table 1). Despite the physically devastating effects of the disease, mental capabilities in RTD remain intact [11].

The time between the onset of deafness and the development of other clinical manifestations varies but commonly occurs within two years. In some individuals, an intercurrent event, usually an injury or infection appears to precipitate the initial manifestations or worsen existing findings [21]. Overall, symptoms are very variable, ranging from severe forms that prevent patients from walking from early childhood to moderate and light forms with the onset in the second decade of life (Table 1).

**Table 1 biomedicines-10-01329-t001:** Clinical features associated with RTD.

General Information		
**Name**	Riboflavin Transporter Deficiency; Brown-Vialetto-Van Laere Syndrome	
**Prevalence**	<1/1,000,000	
**Inheritance**	Autosomal recessive, rarely autosomal dominant	
**Gene Mutations**	*SLC52A2* on chromosome 8q24; *SLC52A3* on chromosome 20p13	
**Cases**	~325 genetically confirmed cases (http://curertd.org/what-is-rtd/history/(accessed on 10 April 2022) Underestimated data [22]	
**Age of Onset**	From few months to early teen years; very rarely adulthood	
**Diagnosis**	Mutational analysis of all genes coding riboflavin transporters;Biochemical tests showing abnormalities such as altered plasma acylcarnitine profiles, abnormal urine organic acids and decreased plasma flavin levels(in 50% of RTD patients)	
**Treatment**	RF supplementation from 10 to 80 mg/kilogram bodyweight/day (400 mg to 2700 mg daily for adults) divided into 2, 3 or 4 doses per day	
**Response to Rf Treatment**	Highly variable, ranging from rapid improvement in days to gradual improvement over several years or stabilization of clinical state.No response is also observed.	
**Clinical Presentation**	Common features include peripheral and cranial neuropathy, neuronal loss in anterior horns and atrophy of spinal sensory tracts, causing muscle weakness, sensory loss, diaphragmatic paralysis and respiratory insufficiency, and multiple cranial nerve deficits such as sensorineural hearing loss, bulbar symptoms, and loss of vision due to optic atrophy	
**Major Manifestations Associated with the Mutated Genes**	*SLC52A2* (*n* = 73)	*SLC52A3* (*n* = 45)
**Auditory Neuropathy**	33%	75% *
**Pontobulbar Palsy**	8%	20% *
**Sensory Gait Ataxia**	64%	5% *
**Optic Atrophy w/o Nnystagmus**	29%	-
**Muscle Weakness**	26%	15% *
**Respiratory Compromise**	8%	10% *

* Percentages calculated on patients diagnosed over 6 years of age.

## 3. Molecular Mechanisms Underlying RTD Pathogenesis

Decreased intracellular levels of FAD and FMN are thought to play a primary role in RTD disease progression. Indeed, these RF derivatives are essential cofactors for numerous dehydrogenases, reductases, and oxidases, mainly located in mitochondria and peroxisomes, being involved in intermediate and terminal energy metabolism of fatty acids, carbohydrates, amino acids, pyridoxine, and choline [17]. Consistently, these organelles are impaired in their integrity and bioenergetics in RTD. Such changes, associated with oxidative stress may also contribute to other cellular alterations, namely cytoskeletal abnormalities. These, in turn, were demonstrated to impact on neuronal differentiation and overall functionality [6,12]

### 3.1. Dysfunctional Metabolic Pathways Depending on Riboflavin and Its Derivatives

It is well known that flavoproteome maintenance, based on RF and flavin cofactor homeostasis, is necessary for cellular energy metabolism. Indeed, flavin cofactors ensure the functionality of fundamental biochemical processes, namely respiratory chain, Krebs cycle, β-oxidation of fatty acyl-CoAs. Inborn errors of such metabolism, caused by disturbances of the coordinated supply of RF and its co-factors, may alter muscular and neuronal flavin-dependent pathways [16]. Besides mitochondrial energy metabolism, other metabolic pathways strongly depend on RF. Indeed, at least 90 genes encoding flavin-dependent proteins are present in our genome [23,24].

Mitochondria and peroxisomes both play major roles in cellular metabolism, especially in lipid biosynthesis/catabolism and ROS production/scavenging. It is now clear that these two organelles metabolically interact, reciprocally regulating their activity [25]. Low levels of RF and its derivatives cause changes in flavoenzyme activities. It is not surprising that the tissues primarily affected by RF homeostasis alterations are the nervous and muscular tissues, since they require higher bioenergetics demand. Even with respect to the cell type, motor neurons and skeletal and heart muscle cells are among the active-most cells, with greatest energy expenditure [16].

#### 3.1.1. Lipid Metabolism

Fatty acids and cholesterol are important building blocks of animal cell membranes, besides being precursors for a wide variety of important biomolecules. Indeed, fatty acids are used in the synthesis of energy-rich triglycerides and signaling molecules, such as prostaglandins. For these reasons, the levels of fatty acids must be tightly controlled, owing to the potentially toxic effects of their accumulation [26].

The role of RF in lipid metabolism, markedly fatty acid oxidation directly correlates with the high flavin content in mitochondria. Both FMN and FAD are transported inside mitochondria by the mitochondrial folate transporter and several studies indicate that the mitochondrion itself is able to produce FAD [24]. Over the past decades, it has become increasingly clear that different metabolic processes depend on the concerted action of peroxisomes and mitochondria. Indeed, β-oxidation occurs in both organelles, thanks to the action of enzymes catalyzing similar chemical reactions, but with largely different substrate specificities [27]. Consistent with the notion that, even in peroxisomes, flavin concentration is prominent, we found reduced expression of peroxisomal proteins involved in fatty acyl β-oxidation pathway [6].

The role of peroxisomes in cellular metabolism is highlighted by the existence of a group of inherited diseases caused by severe impairment of their functions, known as peroxisomal disorders [27,28,29]. In addition, peroxisome alterations have recently been associated with the pathogenesis of diverse neurological disorders [30]. Although peroxisomes are ubiquitous organelles, they greatly vary in their morphology and function, depending on tissue specific features and on metabolic requirements. Thus, their contribution in the functioning of nervous tissue relates to the maintenance of metabolites in the appropriate concentration ranges (for example ROS), as well as in the production of various fundamental brain components. Indeed, peroxisomes are involved in the biosynthesis of phospholipid ethers, or plasmalogens, important constituents of myelin, and polyunsaturated fatty acids (PUFAs), involved in the maintenance of membrane fluidity [31]. Peroxisomes are involved in the last step in the PUFA docosahexaenoic acid (DHA) biosynthesis, known for its role in neurotransmission, synaptic plasticity, and homeostasis of calcium [29]. A lack of either of these functions can cause altered synaptic transmission and/or the onset of inflammatory processes. However, it should be noted that mitochondrial/peroxisomal lipidome is far from being completely understood.

#### 3.1.2. ROS Metabolism

Multiple pathogenetic mechanisms appear to contribute to neuronal degeneration, but strong evidence supports the hypothesis that oxidative stress is an early and crucial trigger [32,33]. Indeed, a fine balance between the presence of ROS and antioxidants is essential for normal cellular signaling, but in case of excessive free radical production or insufficiency of the antioxidant response system, extensive protein oxidation and lipid peroxidation occurs, causing damage to organelles and leading to necrosis or apoptotic cell death [34]. Owing to the high oxygen consumption, elevated content of PUFAs in neuronal membranes and inadequate antioxidant defense, the central nervous system (CNS) is prone to ROS-related damage. Indeed, the effects of oxidative stress within postmitotic cells may be cumulative, and injury by ROS is a major potential cause of neuronal dysfunction, as seen in several neurodegenerative diseases [35,36,37]. To cope with the threat posed by ROS, it is fundamental to maintain redox homeostasis under tight regulation through antioxidant enzymes and low-molecular-weight antioxidants. Among such complex scavenging systems, glutathione reductase (GR) requires FAD for its activity [24]. Consequently, RF deficiency leads to an impaired glutathione-dependent antioxidant defense, presumably increasing lipid peroxidation. In this respect, accumulation of fatty acids -potential targets for lipoperoxidation- due to β-oxidation impairment, may exacerbate oxidative stress.

Peroxisomal FAD or FMN dependent metabolism itself contributes to the generation of ROS and reactive nitrogen species (RNS), as by-products of fatty acyl β-oxidation cycle and other oxidases activities, [38,39]. Since such molecular species participate in controlling signaling pathways especially relevant in excitable cells, the redox balance maintained by peroxisomal antioxidant enzymes, such as catalase, is crucial not only for neuronal survival, but even for proper neurotransmission [40,41,42]. In addition, disturbances in peroxisomal pathways with excess ROS production can result in increased mitochondrial stress and fragmentation [31,38]. Our data on peroxisoms, however, reveal enhanced CAT expression in RTD iPSCs, possibly unveiling response mechanisms aimed at counteracting mitochondrially generated redox imbalance [6]. Such antioxidant response in support of mitochondria may extend to other cytoplasmic compartments, as we reported increased SOD1 expression in RTD cells. On the other hand, decreased SOD2 content may well correlate with the specific mitochondrial involvement in RTD pathomechanisms.

Manole and coll. [21] were the first to report reduced ETC activity in two RTD patients’ fibroblasts. The same Authors showed similar mitochondrial dysfunction in drift knockdown in D. melanogaster brains, accompanied by abnormal mitochondrial ultrastructure, thus accounting for fly locomotor dysfunction. Consistent with these studies, our Focused Ion Beam/Scanning Electron Microscopy (FIB/SEM) analysis of RTD human iPSCs and derived MNs highlighted profound outer and inner mitochondrial membrane abnormalities, with cristae derangement, accompanied by altered mitochondrial membrane potential (MMP) and superoxide anion overproduction [6,43]. Interestingly, the degree of such morphofunctional defects seems to correlate with severity of symptoms of patients, source for iPSCs. In the lack of vertebrate models faithfully recapitulating human RTD, relevant information on mitochondrial involvement in the generation of the pathological in vivo phenotype comes from a recent, innovative study on C. elegans [44]. The Authors produced a metabolic model of flavin deficiency, demonstrating significant impact on mitochondrial bioenergetics, ATP and ROS production in nervous tissue, associated with impaired locomotion. Such study encourages further mechanistic investigation on this simple animal model.

## 4. Cytoskeletal Involvement in RTD

Cytoskeletal derangement, often associated with abnormal protein processing or misfolding and subsequent accumulation is described in various neurodegenerative diseases affecting motor system, including ALS, Charcot-Marie-Tooth disease, giant axonal neuropathy, Parkinson’s disease, diabetic neuropathy, dementia with Lewy bodies and spinal muscular atrophy [45,46].

The main components of the neuronal cytoskeletal network, namely actin microfilaments, microtubules and neurofilaments, are susceptible to oxidation [47]. The relevance of cytoskeleton oxidation depends on the spatiotemporal context in which a defined modification occurs as well as the source of ROS. Whereas physiological ROS production is needed for proper cytoskeletal polymerization, oxidation tends to disrupt polymerization, so that under oxidative stress conditions cytoskeletal dynamics is impaired [48,49].

Recently, different studies carried out on RTD iPSC and derived MNs highlighted the presence of cytoskeletal aberrations in RTD iPSC and derived MN-specific patient cells [6,12,50]. Specifically, we observed abnormal colony-forming ability and loss of cell–cell contacts in patients’ iPSCs, by light, electron, and confocal microscopy, using tight junction marker ZO-1 [6], whereas Rizzo and coll. [50] demonstrated perturbation in neurofilament composition of RTD MNs, suggesting impairment in axonogenesis and neurotrophic factor metabolism. Indeed, the Authors showed downregulation of genes related to neurite outgrowth and guidance (neuronal pentraxins, palladin, and ephrin-A2) in RTD MNs.

On the other hand, Niceforo and coll. [12] demonstrated abnormal expression and distribution of α- and β-tubulin, as well as altered tyrosination of α-tubulin, accompanied by an impaired ability to re-polymerize after nocodazole treatment in RTD patient-derived iPSCs. Consistent changes in tubulin content associated with abnormal morphofunctional features, such as neurite length and Ca^2+^ homeostasis in RTD MNs, thus supporting impaired differentiation [12].

Alterations in cytoskeletal structure and arrangement well correlates to evidence regarding oxidative stress in RTD, supporting the interactive involvement of the two processes in the pathogenesis of RTD [6,43,49,50,51,52,53].

## 5. In Vivo and In Vitro Models of RTD

As for other rare diseases, even for RTD in vivo models faithfully recapitulating clinical symptoms and progression of pathology are lacking. Perinatal lethality of *Slc52a3*^−/−^ mouse strain, due to severe metabolic disorder [54] prevented researchers from addressing whether the gene has an important function in neurodevelopment or short-term in vitro survival of motor neurons from the spinal cord. To address this issue, Intoh and coll. [54] differentiated mouse embryonic stem (ES) cell lines with *Slc52a3*^+/+^, *Slc52a3*^+/−^ and *Slc52a3*^−/−^ genotypes into motor neurons demonstrating that, even in the normal genotype, *Slc52a3* expression was significantly and dramatically reduced after 7 days of differentiation and essentially absent at 14 days when compared with ES cell levels. Therefore, *Slc52a3* is unlikely to play an important role in the development of spinal motor neurons, at least in mouse [54]. Independently, Yoshimatsu et al. [55] reported that *Slc52a3*^−/−^ mice die with hyperlipidemia and hypoglycemia within 48 h after birth, and that organogenesis in these mice is apparently normal, although their body weight is significantly lower. The Authors hypothesized that such metabolic defects are due to impaired fatty acid oxidation, which is consistent with the knowledge that defects in mitochondrial β-oxidation result in mouse neonatal mortality [56]. Even though unsuited to recapitulate RTD late neurodevelopmental pathology, these models have brought to light the importance of RF transport during embryonic development, an issue which also deserves further investigation.

Moreover, Subramanian and coll. [57] developed a conditional (intestinal-specific) RFVT-3 knockout (cKO) mouse model by using a Cre/Lox approach. RFVT-3 cKO mice showed significant growth retardation and died between the age of 6 and 12 weeks. Additionally, cKO animals exhibited lower body weights, lethargic behavior, hunched back posture, and ocular surface abnormalities, compared with WT littermates [57].

In a more recent study, the same research group demonstrated abnormal development of the cerebral cortex in *Slc52a3*^−/−^ mice, as a consequence of lower RF levels [58].

Using D. melanogaster as an in vivo model system, Manole and coll. [21] demonstrated that knockdown of drift, i.e., the fly RF transporter homologue, causes a severe impairment in locomotor activity and reduced lifespan, mirroring patient pathology. These features were associated with profound neuronal abnormalities, particularly involving mitochondrial ultrastructure [21].

As an alternative in vivo approach, C. elegans was recently proposed as a low-cost and sustainable model to elucidate the molecular rationale for RF therapy in human RF responsive neuromuscular disorders. In order to mimic RTD pathology, *flad-1*, orthologue of human flavin adenine dinucleotide synthetase (FLAD1), was silenced in a model strain hypersensitive to RNA interference in nervous system [44]. As a result, altered flavin homeostasis, with impact on mitochondrial bioenergetics and ROS production, was observed, in association with altered behavioral patterns [44]. A pre-synaptic neuronal defect linked to FAD decrease and secondary to acetylcholine deficit was suggested to underlie the locomotion phenotype of *flad-1*-silenced animals (Figure 2).

A rescue of the phenotype induced by *flad-1* silencing was observed following exogenous flavins treatment, indicating that increased availability of the coenzymes could have a chaperoning effect on endogenous flavoproteins [44,59].

This evidence is consistent with data collected by our research group using a patient-derived model of RTD, based on induced pluripotent stem cell (iPSC) technology. In fact, primary involvement of mitochondrial and peroxisomal-related energy dysmetabolism in RTD was demonstrated. Mitochondrial ultrastructural abnormalities, redox imbalance, altered expression of antioxidant systems, and peroxisomal deregulation were demonstrated in RTD patient-specific iPSCs and derived motor neurons (MNs) [6,43]. The involvement of the above organelles is not surprising, since the availability of RF and its cofactors is essential for correct functioning of flavoenzyme-dependent pathways. Mitochondrial and peroxisomal alterations are also related to an altered redox state caused by an imbalance of ROS production vs. antioxidant response, which also may affect cytoskeletal structure, as we observed in patient-derived iPSCs [10]. Notably, ROS were suggested to modulate microtubule dynamics in cells [51] (Table 2).

In this context, it should be mentioned that is now possible to simultaneously generate spinal cord neurons and skeletal muscle cells that self-organize to generate human neuromuscular organoids that can be maintained in 3D for several months [60]. Indeed, 3D organoids mimic tissue cytoarchitecture and functionality, allowing multiple applications, including drug screening and regenerative medicine [40,61,62]. Thus, developing RTD organoids would provide an excellent tool for understanding the complex pathogenic cell processes, while maintaining the accessibility typical of an in vitro system.

## 6. Therapeutic Prospects for RTD

Empirical studies showed that RF administration gradually improves symptoms, as muscle strength, motor function, respiration, hearing, and vision in some children affected by RTD [64,65,66]. This prompted researchers to investigate at the molecular and cellular level possible beneficial effects of RF supplementation to RTD in vivo or in vitro models.

Yoshimatsu and coll. [55] examined the effect of RF supplementation to *Slc52a3*^−/−^ mice, finding that survival of *Slc52a3*^−/−^ pups was comparable to that of WT littermates. The Authors also found plasma RF levels significantly increased in *Slc52a3*^−/−^ mice when their dams were given RF via drinking water. Moreover, weight loss at birth, analysis of plasma acylcarnitine and glucose levels in treated *Slc52a3*^−/−^ neonatal mice improved by RF supplementation [55].

Additionally, Manole and coll. [21] asked whether locomotor defects in *drift* knockdown flies could be rescued by RF supplementation. They demonstrated that esterified RF derivative rescues *drift* knockdown phenotypes. Specifically, vitamin supplementation resulted in heightened locomotion and extension of lifespan in *drift* knockdown flies. This rescue correlated with substantially increased complex I activity [21].

Remarkably, the translational potential of the use of patient-specific cells is ensured by the readily transferable findings to preclinical studies. In this context the iPSC model, thanks to its ability to differentiate in vitro into the disease-affected cell type, offers the unique opportunity to test innovative drug treatments on the patient’s own cells, while elucidating the pathogenetic processes of neurological diseases. Indeed, RF partially rescues cell phenotype, neurofilament expression pattern, and redox status, which was associated with improving ultrastructural features of mitochondria in RTD iPSCs and derived neurons [6,43]. This strongly supports RF treatment to restore mitochondrial- and peroxisomal-related aspects of energy dysmetabolism and cope with oxidative stress in RTD [6]. Moreover, RF administration partially improved cytoskeletal features in patients’ iPSCs and MNs, suggesting that the redundancy of the transporters may rescue cell functionality in the presence of adequate concentrations of the vitamin [12,50]. Alternative pathways of RF absorption and transport to the CNS, however, require further studies aimed at clarifying whether RFT isoforms are differentially expressed in neural cell types (neurons, endothelial and astroglial cells), or subtypes (motor neurons vs. other neuronal populations). On the other hand, unfortunately, improvements are not observed in all RTD patients treated with excess RF, so raising the possibility that non-responder individuals are unable to activate different means of RF transport and emphasizing the need for more specific treatments.

It is worth mentioning that the relationship between RF intracellular scarcity and biochemical damage to neurons, as well as the molecular mechanisms regulating flavin transport efficiency in different types of neurons, is yet to be clarified [1].

Most human pathologies linked to enzymatic cofactor deficiency can be treated with high doses of vitamin B2, but the reasons why some patients ameliorate, and others do not, still need to be elucidated.

Thus, considering the susceptibility of neurons to redox imbalance and the role played by oxidative stress in the onset and progression of neurodegenerative disorders, molecules with antioxidant properties are receiving increasing attention for the treatment or prevention of these pathologies [36]. For this reason, studies on RTD iPSCs and derived MNs focused on the effects of the following antioxidants: N-acetyl cysteine (NAC), Vitamin C, Idebenone, Coenzyme Q10 and EPI-743 [12,43,52]. Among these molecules, whose beneficial effects have already been extensively elucidated in other models of neurodegenerative diseases [67,68,69], NAC restored mitochondrial morphology and functionality in RTD MNs [43]. It is also worth dwelling on the role of EPI-743, a synthetic, vitamin E-like para-benzoquinone antioxidant, which readily crosses the blood–brain barrier, increasing glutathione biosynthesis [70,71]. EPI-743 is a fat-soluble compound already used for some metabolic diseases, e.g., Leigh syndrome, a rare, fatal inherited neurodegenerative disorder [72]. The effects of EPI-743 were the most satisfactory in terms of restoring the altered phenotype. Compared to other antioxidant molecules, EPI-743 likely exerts a more complex action, since it modulates multiple cell pathways inducing nuclear factor erythroid 2-related factor 2 (Nrf2), an important regulator of cellular resistance to ROS damage [52,73,74].

## 7. Perspective

In this review, we addressed the pathogenic mechanisms of RTD, focusing on the major players in the disease, i.e., energy dysmetabolism and cytoskeletal derangement. Indeed, redox imbalance, due to inadequate levels of intracellular RF, affects cytoskeletal integrity. Flavoenzyme inefficiency leads to disturbances in reactions of intermediate metabolism. It is worth mentioning that defective lipid β-oxidation not only causes ATP depletion but has a wide array of cellular and possibly systemic implications.

Even more relevant for MN functionality, low acetyl CoA levels, resulting from impaired mitochondrial and peroxisomal β-oxidation pathways, likely determine reduced acetyl choline synthesis by choline acetyl transferase. Such defective neurotransmission was recently suggested by Barile’s group [44], based on their innovative RF deficiency *C. elegans* model. To this respect, the relatively high expression levels of acyl-CoA oxidase observed in rodent cranial and spinal MNs, particularly in axons, argues for a specific involvement of peroxisomes in cholinergic neurotransmission [75]. This notion could partly explain the selective vulnerability of MNs. Further studies exploring this issue and aimed at identifying possible targets to ameliorate such defects are needed. Along with the pre-synaptic neuronal defect, considered as *primum movens* of RTD, muscle tissue pathogenesis should not be underrated, confining it to a merely passive role. In the next future, researchers should address the specific and primary damage to energy metabolism pathways and cytoskeletal dynamics in muscle cells, to ascertain the relative contribution to locomotor dysfunction and the neuron-muscle pathological crosstalk occurring in RTD condition. Notably, ALS, a much more common MN disease, is recently being approached by a wider perspective, taking into account implication of non-neural tissues, markedly, but not limited to, muscle, in the onset and progression of the disorder [76].

While waiting for a suitable in vivo model to study multisystemic impact of RFT mutations, patient specific iPSCs appear the most reliable and promising in vitro model. Thanks to this technology, the knowledge on RTD has increased, emphasizing the relationship between oxidative stress and cytoskeletal derangement. Such connection may explain the complex cellular pathophenotype, in terms of impaired organelle function and transport, altered cell–cell communication and neurotransmitter deprivation. Among other, secondarily affected, cell processes, apoptosis and autophagy are most likely to be disturbed, leading to general cell homeostasis deregulation and death [77]. Special attention should be given to mitophagy and pexophagy, whose specific neuroprotective role in RTD has only been partially addressed [50].

Moreover, the existence of a strictly causal relationship between each genetic mutation and the clinical phenotype is yet to be addressed. In this context, CRISPR-Cas9 technology has opened the possibility of directly editing genomic sequences, elucidating the contribution of genetics to disease, by promoting the creation of more accurate cellular and animal models. Combination of iPSC technology with CRISPR-Cas9 gene editing has launched a new era in gene therapy for the treatment of neurological disorders [78,79].

An especially critical therapeutic aspect concerns the timing of RF administration. Indeed, high-dose RF treatment ameliorates the progression of the disease, when initiated soon after the onset of symptoms. To stabilize clinical signs, it should therefore start as soon as RTD is suspected and be continued lifelong after diagnosis confirmation [80].

Overall, novel therapeutic strategies, based on the combination of RF, antioxidants and possibly other neuroprotective molecules acting on specific survival, differentiation, and functional pathways, involving specific organelles, are to be designed for RTD treatment (Figure 3).

## Figures and Tables

**Figure 1 biomedicines-10-01329-f001:**
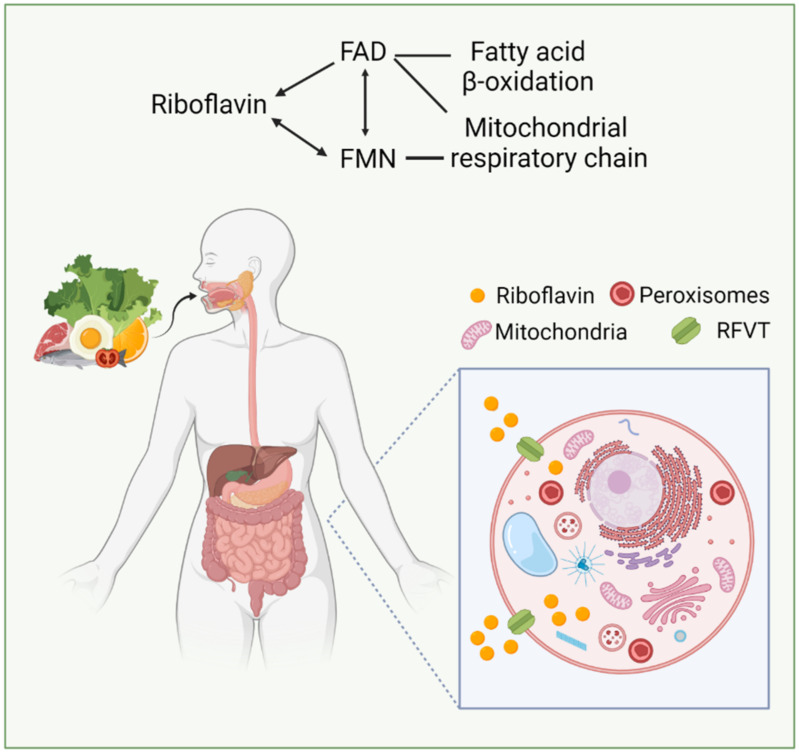
Physiological RF uptake and transport. RF can be found in a wide variety of natural sources, especially meats, eggs, fish, some fruits and legumes, wild rice, vegetables, cheese, and dietary products. The majority of RF is absorbed in the small intestine, and, to a lesser extent in the stomach, duodenum, colon and rectum. RF absorption and transport depend on a group of transporters from the solute carrier family *SLC52* (RFVT) [20]. RF is the precursor of FMN and FAD (arrows), which in turn are involved in fatty acid β-oxidation and mitochondrial respiratoy chain (connectors). The figure was created with BioRender.com (accessed on 13 April 2022).

**Figure 2 biomedicines-10-01329-f002:**
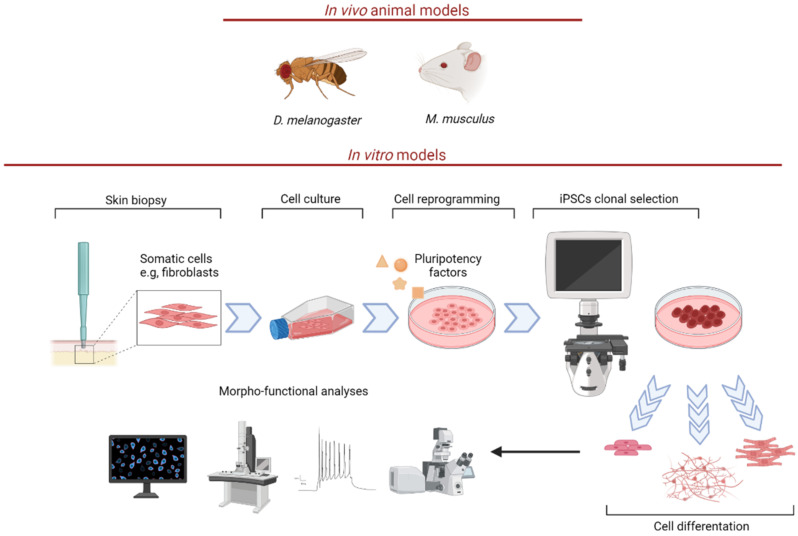
RTD in vivo and in vitro models. Proper RTD animal models include M. musculus and D. melanogaster. Note that C. elegans was also used to mimic human RF metabolism deficiency. In vitro models include patient-specific fibroblasts and iPSCs, which can be differentiated into MNs. Indeed, skin biopsy-derived fibroblasts can be reprogrammed into iPSCs, thanks to the introduction of specific pluripotency factors. Human iPSCs may be differentiated into cell types. In the figure, as an example intestinal epithelium, muscle, and neurons, as representative derivatives from endoderm, mesoderm and ectoderm germ layers, are illustrated. Neurons may be studied using diverse morphofunctional approches (e.g., confocal and electron microscopy, electrophysiology) The figure was created with BioRender.com (accessed on 13 April 2022).

**Figure 3 biomedicines-10-01329-f003:**
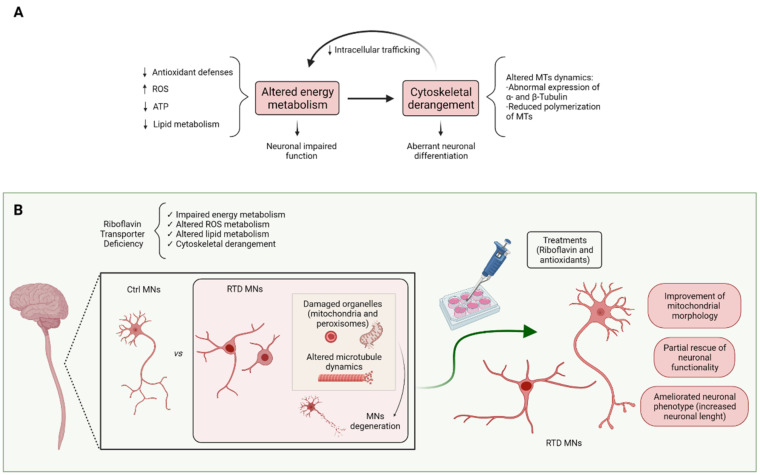
(**A**). As a precursor of FMN and FAD, RF plays a crucial role in energy metabolism. Deficiency of RF transporters leads to mitochondrial and peroxisomal energy dysmetabolism and altered redox state which also affects cytoskeletal arrangements (e.g., altered MTs dynamics). Arrows indicate increased or decreased concentration of the specific molecules. (**B**). RTD-patient-derived MNs exhibit altered morphology associated with ROS overproduction. Diseased cells also show damaged organelles and MN degeneration. RF treatments alone or combined with other antioxidants result in a partial rescue of the altered phenotype. MTs, microtubules; MNs, motor neurons; ↑ and ↓ indicate increased or decreased levels. The figure was created with BioRender.com (accessed on 13 April 2022).

**Table 2 biomedicines-10-01329-t002:** RTD in vivo and in vitro models.

RTD Models	
** *D. melanogaster* **	Drift knockdown results in abnormal mitochondrial morphology, reduced ETC activity, and lower MMP [21].
** *M. musculus* **	*Slc52a3*^+/−^ mice display same body weight and plasma RF concentration as in WT. Intercrossing between *Slc52a2*^+/−^ mice fail to generate *Slc52a2*^−/−^ mice, due to early embryonic lethality, for a failure in placental development [54]. *Slc52a3*^−/−^ mice die from hyperlipidemia and hypoglycemia within 48 h after birth [55]. They show abnormal development of the cerebral cortex and altered brain morphology [55]. *Slc52a3* conditional KO in the intestine also leads to early postnatal death (6–12 weeks) [57].
** *Fibroblasts* **	Skin fibroblasts from two RTD patients carrying either *SLC52A2* or *SLC52A3* mutations show significant reduction in RF transport [63]. Skin fibroblasts from patients with *SLC52A2* mutations reveal significant reduction in ETC complex I and II activity [21].
** *iPSCs and MNs* **	iPSCs-derived MNs from RTD patients carrying *SLC52A3* gene mutations show cytoskeletal perturbation [50]. iPSCs and iPSCs-derived MNs from RTD patients carrying *SLC52A2* gene mutations show altered redox balance together with cytoskeletal derangement [6,12,43].

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
