# Peer review of "New Insights into the Neurodegeneration Mechanisms Underlying Riboflavin Transporter Deficiency (RTD): Involvement of Energy Dysmetabolism and Cytoskeletal Derangement"

_biomedicines, 2022, doi:10.3390/biomedicines10061329_

Round 1
Reviewer 1 Report
In this review article, the authors present an overview of a rare genetic disorder caused by inactivating mutations in the SLC52A2 and SLC52A3 genes encoding riboflavin (RF, vitamin B2) transporter. Known as riboflavin transporter deficiency (RTD), this disorder is a progressive neurodegenerative disease characterized by cranial and peripheral neuropathies that lead to breathing difficulties, facial weakness, hearing loss, abnormal eye movements, difficulty chewing and swallowing, muscle weakness of the arms and legs, and an unsteady or unbalanced gait. The authors discuss the mechanisms by which RTD impacts the metabolic pathways dependent on riboflavin and its derivatives, flavin mononucleotide (FMN) and flavin adenine dinucleotide (FAD), and alter fatty acyl ß-oxidation, mitochondrial oxidative phosphorylation and reactive oxygen species production as well as cytoskeletal oxidation in motor and sensory neurons. Because RTF mouse models exhibit perinatal lethality, other in vivo and in vitro models have been studied, and models based on patient-derived induced pluripotent stem cells (iPSCs) have been found to replicate, at least in part, altered redox balance and/or cytoskeletal derangement. Therapeutic supplementation with riboflavin has produced only partial amelioration of RTD patients, and in iPSC models, treatment with antioxidant agents appear to yield additional benefits. The authors conclude by mentioning the need for further research in areas such as variable symptomatology and reduced synthesis of acetylcholine neurotransmitter.
This review article could be an interesting contribution given its focus on the molecular pathogenesis of RTD and the potential utility of predictive RTD models, such as iPSCs in identifying better therapeutic approaches. However, this article suffers from a major issue, which is that the flow of ideas simply does not flow easily in a logical manner throughout the entire manuscript. Here are some examples.
The abstract does not reflect the content of the article. It is not enough to describe the in vivo and in vitro models, including iPSCs. Make the abstract accessible to a broad audience by first describing the genetic and molecular bases of the metabolic and cytoskeletal derangements in RTD and their association with the clinical manifestations and current therapy of this disease before dwelling into models of disease and their applications.
Similarly, in the introduction, the first paragraph about iPSC should be moved to the end of that section.
Section 3 about in vivo and in vitro models of RTD should be moved to after current section 5 about cytoskeletal involvement in RTD.
Figure 1 should be shown with section 2 about the pathophysiology of riboflavin transport and RTD.
Overall, the entire manuscript must be revised extensively to improve its clarity, conciseness and impact.
Author Response
Reply to Review Reports_Ms. 1705491
Reviewer 1
Comments and Suggestions for Authors: In this review article, the authors present an overview of a rare genetic disorder caused by inactivating mutations in the SLC52A2 and SLC52A3 genes encoding riboflavin (RF, vitamin B2) transporter. Known as riboflavin transporter deficiency (RTD), this disorder is a progressive neurodegenerative disease characterized by cranial and peripheral neuropathies that lead to breathing difficulties, facial weakness, hearing loss, abnormal eye movements, difficulty chewing and swallowing, muscle weakness of the arms and legs, and an unsteady or unbalanced gait. The authors discuss the mechanisms by which RTD impacts the metabolic pathways dependent on riboflavin and its derivatives, flavin mononucleotide (FMN) and flavin adenine dinucleotide (FAD), and alter fatty acyl ß-oxidation, mitochondrial oxidative phosphorylation and reactive oxygen species production as well as cytoskeletal oxidation in motor and sensory neurons. Because RTF mouse models exhibit perinatal lethality, other in vivo and in vitro models have been studied, and models based on patient-derived induced pluripotent stem cells (iPSCs) have been found to replicate, at least in part, altered redox balance and/or cytoskeletal derangement. Therapeutic supplementation with riboflavin has produced only partial amelioration of RTD patients, and in iPSC models, treatment with antioxidant agents appear to yield additional benefits. The authors conclude by mentioning the need for further research in areas such as variable symptomatology and reduced synthesis of acetylcholine neurotransmitter. This review article could be an interesting contribution given its focus on the molecular pathogenesis of RTD and the potential utility of predictive RTD models, such as iPSCs in identifying better therapeutic approaches. However, this article suffers from a major issue, which is that the flow of ideas simply does not flow easily in a logical manner throughout the entire manuscript. Here are some examples.
-The abstract does not reflect the content of the article. It is not enough to describe the in vivo and in vitro models, including iPSCs. Make the abstract accessible to a broad audience by first describing the genetic and molecular bases of the metabolic and cytoskeletal derangements in RTD and their association with the clinical manifestations and current therapy of this disease before dwelling into models of disease and their applications.
We thank the Reviewer for this relevant comment. In the revised version of the manuscript, we modified the Abstract accordingly, adding details on RTD pathology and on the contribution of metabolic and cytoskeletal derangements to the pathogenesis. We feel that now the Abstract reflects the content of the article.
-Similarly, in the introduction, the first paragraph about iPSC should be moved to the end of that section.
We agree with the Reviewer, so we moved the paragraph about iPSC technology to the end of the section.
-Section 3 about in vivo and in vitro models of RTD should be moved to after current section 5 about cytoskeletal involvement in RTD.
We thank the Reviewer for this comment. However, we feel that the flow of information given by the review article would be compromised by such a change. Indeed, to dwell into the pathogenic mechanisms which have been dissected taking advantage of the different in vivo and in vitro models, it is necessary to introduce such models. In this way, pathogenic mechanisms may be discussed in an integrated manner, independent of the specific model used from time to time. We wish here to also reply to the main criticism of the Reviewer (“However, this article suffers from a major issue, which is that the flow of ideas simply does not flow easily in a logical manner throughout the entire manuscript.”), clarifying that the general aim of our manuscript is to discuss data from different Authors working on different RTD models, so to provide an overview of the RTD pathomechanisms, which may in turn help identifying novel therapeutic targets for the disease.
-Figure 1 should be shown with section 2 about the pathophysiology of riboflavin transport and RTD.
We thank the Reviewer for this point. It is true that former Figure 1 contained information related to sections 2 and 3, in its parts (a) and (b). Therefore, in the revised manuscript, we split the figure in three, so that Figure 1 now contains former (a), while former (b) part is moved to Figure 2, which also contains an additional panel. Former part (c) is now an independent Figure (3), with further details, at the end of the Conclusions.
-Overall, the entire manuscript must be revised extensively to improve its clarity, conciseness, and impact.
We thank the Reviewer for this suggestion. We revised our manuscript hoping to have improved the clarity. All the changes we made are highlighted in yellow.

Reviewer 2 Report
This is an interesting and timely review about findings in recent years using various types of RTD models. The authors have expertise in the field and have provided insightful mechanistic views on the disease. I have concerns in the following respects:
1) Illustrations
The authors stated that this article is focused on the energy dysmetabolism and cytoskeleton derangement involved in RTD. Therefore to include schematic drawings to illustrate the two aspects would aid readers' understanding, thereby enhancing their accessibility to the article.
2) Citations for evidence
It would be better to cite evidence to support the following statements :
On page 2 (lines 61 and 62), "Both organelles are also a major source of reactive oxygen species (ROS), whose release contribute to neuronal death."
On page 7 (lines 249-251), "Multiple pathogenetic mechanisms appear to contribute to neuronal degeneration, but strong evidence supports the hypothesis that oxidative stress is an early and crucial trigger."
3) English writing
There are many lengthy sentences in this article. Those sentences would cause readers' confusion. One example about this is on page 9, the sentence, "The ineffectiveness of the RF treatment on some patients, is leading researchers on the one hand to further explore alternative therapeutic strategy, and on the other hand, to investigate molecular mechanisms underlining RF absorption and transport." It would be helpful to revise by simplifying the compound sentence to simple ones for passing clearer messages.
On page 10 (line 437), the authors used the word "suspect". Is "possibility" a meaningful replacement?
There are punctuation and grammatical errors in the article.
For example, on page 2 (line 49), "fatty acid -oxidation"; one page 2 (line 62) "contribute" ("contributes"?); on page 8 (line 300), "includingALS".
I fully appreciate that the first language for the authors is not English; therefore, in this regard the authors should seek help from professional(s) or from a native speaker of English.
4) Formatting
The authors used "colleagues" in most places, however, on page 8 (lines 333 and 339), "coll." was used twice. It would be better to use an abbreviation consistently.
The citation format does not seem to be consistent. For example, there are two styles: before page 9, citations formatted, "[7], [55], [57]", and page 9 onward, the style changed to "[10, 25, 56]".
The phrases, "in vitro" and "in vivo", are not always in italic font.
It would be better to place the table 1 on the same page.
Author Response
Reply to Review Reports_Ms. 1705491
Reviewer 2
Comments and Suggestions for Authors: This is an interesting and timely review about findings in recent years using various types of RTD models. The authors have expertise in the field and have provided insightful mechanistic views on the disease. I have concerns in the following respects:
1) Illustrations: The authors stated that this article is focused on the energy dysmetabolism and cytoskeleton derangement involved in RTD. Therefore, to include schematic drawings to illustrate the two aspects would aid readers' understanding, thereby enhancing their accessibility to the article.
We absolutely agree with the Reviewer and modified the illustrations according to his/her suggestion. Now the manuscript includes three figures, with details in Figure 3 on the relationship linking energy dysmetabolism and cytoskeleton derangement in RTD.
2) Citations for evidence: It would be better to cite evidence to support the following statements:
-On page 2 (lines 61 and 62), "Both organelles are also a major source of reactive oxygen species (ROS), whose release contribute to neuronal death."
We thank the Reviewer for the suggestion. We have added the following citations to support the statements: Valencia A and Morán J. Reactive oxygen species induce different cell death mechanisms in cultured neurons. Free Radic Biol Med. 2004;36(9):1112-1125; Boldyrev A, et al. Neuronal cell death and reactive oxygen species. Cell Mol Neurobiol. 2000;20(4):433-450; Redza-Dutordoir M, and Averill-Bates DA. Activation of apoptosis signalling pathways by reactive oxygen species. Biochim Biophys Acta. 2016;1863(12):2977-2992)
-On page 7 (lines 249-251), "Multiple pathogenetic mechanisms appear to contribute to neuronal degeneration, but strong evidence supports the hypothesis that oxidative stress is an early and crucial trigger."
We thank the reviewer for the suggestion. We have added the following citations to support the statement: Gandhi S and Abramov AY. Mechanism of oxidative stress in neurodegeneration. Oxid Med Cell Longev. 2012; 2012:428010; Kim GH, et al. The Role of Oxidative Stress in Neurodegenerative Diseases. Exp Neurobiol. 2015;24(4):325-340.
3) English writing: There are many lengthy sentences in this article. Those sentences would cause readers' confusion.
-One example about this is on page 9, the sentence, "The ineffectiveness of the RF treatment on some patients, is leading researchers on the one hand to further explore alternative therapeutic strategy, and on the other hand, to investigate molecular mechanisms underlining RF absorption and transport." It would be helpful to revise by simplifying the compound sentence to simple ones for passing clearer messages.
We thank the reviewer for the suggestion. In the revised version of the manuscript, we simplified the confused sentence.
-On page 10 (line 437), the authors used the word "suspect". Is "possibility" a meaningful replacement?
Thank for the suggestion, we replaced the word “suspect” with “possibility” in the revised version of the review.
-There are punctuation and grammatical errors in the article.
For example, on page 2 (line 49), "fatty acid -oxidation"; one page 2 (line 62) "contribute" ("contributes"?); on page 8 (line 300), "includingALS".
We have checked grammar, punctuation and writing mistakes in the revised version of the manuscript.
-I fully appreciate that the first language for the authors is not English; therefore, in this regard the authors should seek help from professional(s) or from a native speaker of English. We thank the reviewer for the suggestion. We have checked the English language with the help of a mother-tongue.
4) Formatting: The authors used "colleagues" in most places, however, on page 8 (lines 333 and 339), "coll." was used twice. It would be better to use an abbreviation consistently.
We replaced “colleagues” with the abbreviation form “coll.” in the revised version of the manuscript.
-The citation format does not seem to be consistent. For example, there are two styles: before page 9, citations formatted, "[7], [55], [57]", and page 9 onward, the style changed to "[10, 25, 56]".
We thank the reviewer for this point. In the new version of the paper, we have standardized the style of the cited references in the text.
-The phrases, "in vitro" and "in vivo", are not always in italic font.
We have checked the italic font in the phrases “in vivo” and “in vitro”.
-It would be better to place the table 1 on the same page.
We agree with the reviewer and in the new version of the manuscript, we have placed the Table 1 on the same page.

Round 2
Reviewer 1 Report
I have reviewed the authors' response letter and revised article with the anticipation of a much improved manuscript. Although the authors did revise some of the text and figures (highlighted in yellow), I strongly believe that they did not adequately address my main concern, which is the clarity of the flow of ideas. Another concern is the description and comparison in quantitative terms of the impact of the many mechanisms of disease implicated in Riboflavin Transporter Deficiency. Throughout the manuscript, the authors list many biological functions that are increased or decreased without detailed information about the reference group and the magnitude, the time course and the potential interactions of these changes. Such listings are pretty much useless to advancing our understanding of the relative impact of the different mechanisms of disease and in identifying better therapeutic target(s). The revisions made by the authors have been minimal and nowhere near the extent needed to make the manuscript a strong, original critique of the literature and contribution to knowledge.
Author Response
Dear Reviewer,I wish to thank you for your time spent in evaluating our manuscript and for your useful suggestions.Specifically, in this second round of revision, we tried to meet all your requests, concerning the flow of ideas in the manuscript.
As you may see from the revised version of the manuscript, we moved concepts and related figures, from one paragraph to another, modifying the overall structure of the manuscript. Please note that all the changes we made are now in “revision mode”.We hope that our review article is now suitable for publication in Biomedicines Special Issue.Sincerely,
Sandra Moreno
Reviewer 2 Report
In my view, the manuscript quality has been substantially improved through this revision. The inclusion of the two new figures should increase the accessibility for readers in scientific communities and attach great attention from lay audience, thereby enhancing the impact of this work upon its publication. Moreover, new citations that have been included have strengthened the authors' scientific points. Furthermore, the scientific writing and format have been improved in the current version.
Author Response
Dear Reviewer,
I wish to thank you for your appreciation, also on behalf of my colleagues. It was a great experience working with your cooperation. We feel that the paper improved a lot, thanks to your suggestions.
Best regards,
Sandra Moreno
Round 3
Reviewer 1 Report
In the latest version of their review article, the authors have improved the flow of ideas by reorganizing several sections of the manuscript.
Under keywords, please write acronyms RTD and iPSC in full.
Line 82-83: what are some of the most common mutations (SNP, indel, fusion, etc.) of SLC52A2 and SLC52A3 genes?
Line 86: after “placental transport defects”, provide reference(s).
Fig. 1: what is the direction of arrows connecting FAD and FMN to fatty acid β-oxidation and mitochondrial respiratory chain?
Table 1: replace “major symptoms based on the mutated gene” by “major manifestations associated with the mutated genes”. Replace “Pontobulbar involvement” by “Pontobulbar palsy”. Add typical time of onset in separate columns.
Line 210: what does “RNS” stand for?
Line 263: what is “imbalanced tyrosination of α-tubulin”?
Fig. 2: C. elegans is a member of Kingdom Animalia; therefore, it is an animal model. Under Cell Differentiation, indicate which cell types are illustrated.
Table 2: what is the tissue source (skin?) of the fibroblasts, iPSCs and MNs? Under Fibroblasts, is it “RTD patients carrying either SLC52A2 or SLC52A3 mutations”?
Line 444: which symptoms are improved or not by riboflavin administration?
Line 480-482: summarize in 1-2 sentences the studies about antioxidants in RTD iPSCs and derived MNs.
Line 493: this section should be retitled “Perspective”. It should also be more concise and to the point.
Finally, the authors did not address the question raised about including quantitative information when describing and comparing the mechanisms of disease implicated in Riboflavin Transporter Deficiency. At a minimum, the authors must indicate which effects are major effects or minor effects, preferably with supporting numbers and statistics, so that the readers can better understand the biological significance of these mechanisms.
Author Response
Dear Editor,
We thank you for having taken into consideration for publication our manuscript entitled “New insights into the neurodegeneration mechanisms underlying Riboflavin Transporter Deficiency (RTD): involvement of energy dysmetabolism and cytoskeletal derangement”, by Fiorella Colasuonno, Chiara Marioli, Marco Tartaglia, Enrico Bertini, Claudia Compagnucci and myself.
We also wish to thank the Reviewer for his/her further suggestions, according to which we modified the text and figures. We hereby enclose a point-by-point response to the Reviewer’s comments.
To facilitate re-revision of the manuscript, all changes have been marked up using the “Track Changes” function or been highlighted (in yellow color).
We hope that our manuscript in its revised version is now suitable for publication in Biomedicines.
Best regards,
Sandra Moreno
Reply to Reviewer 1
Comments and Suggestions for Authors:
-In the latest version of their review article, the authors have improved the flow of ideas by reorganizing several sections of the manuscript.
We thank the Reviewer for appreciation.
Under keywords, please write acronyms RTD and iPSC in full.
Following the Reviewer’s suggestion, in the revised version of the article, we wrote Riboflavin Transporter Deficiency and induced pluripotent stem cells in full, under keywords.
-Line 82-83: what are some of the most common mutations (SNP, indel, fusion, etc.) of SLC52A2 and SLC52A3 genes?
We thank the Reviewer for this point. The most common mutations of SLC52A2-3 genes are homozygous nonsense and missense mutations. Beside the article by Green and coll. (2010) [Ref 14 in our manuscript] in which the Authors described in 9 RTD patients carrying nonsense and missense mutations, very recently, Jin and Yonezawa [Ref 15 in the revised version of the manuscript] published a review in which pathogenic variants of RFVT1/SLC52A1, RFVT2/SLC52A2 and RFVT3/SLC52A3 are listed and described. They also include intron variant, splice site, frameshift and synonymous. In the revised version of the article, we clarified this point.
-Line 86: after “placental transport defects”, provide reference(s).
In the revised version of the manuscript, we added references to support this sentence. This point is also discussed in paragraph 5 “In vivo and in vitro models of RTD” (Refs. 54 and 58)
-Fig. 1: what is the direction of arrows connecting FAD and FMN to fatty acid β-oxidation and mitochondrial respiratory chain?
We understand your point, but we did not mean to represent a biochemical process (as in RF -->FMN-->FAD), but rather the involvement of cofactors in the cited functions. Therefore, we left connectors, the meaning of which is now explained in Fig. 1 legend.
-Table 1: replace “major symptoms based on the mutated gene” by “major manifestations associated with the mutated genes”. Replace “Pontobulbar involvement” by “Pontobulbar palsy”. Add typical time of onset in separate columns.
We thank the Reviewer for this comment. In the revised version of the manuscript, we replaced the sentences with the suggested form. However, it was not possible to add the requested columns, for lack of sufficient information on time and onset of symptoms.
-Line 210: what does “RNS” stand for?
We wrote “reactive nitrogen species”, in place of acronym RNS.
-Line 263: what is “imbalanced tyrosination of α-tubulin”?
With this sentence we referred to experiments described in the article by Niceforo and coll. (2021) (Ref 12) in which the Authors, by studying the tyrosination/detyrosination of alpha tubulin cycle, suggested instability of microtubules in RTD iPSCs. We replaced “imbalanced” with “altered”, for the sake of clarity.
-Fig. 2: C. elegans is a member of Kingdom Animalia; therefore, it is an animal model. Under Cell Differentiation, indicate which cell types are illustrated.
We thank the Reviewer for the suggestions about Figure 2. The reason why we cannot include C. elegans among the animal models showed in Figure 2, is that it does not represent a proper genetic RTD model, since there are no alterations of genes encoding RF transporters. Indeed, this member of Kingdom Animalia has been used to mimic human riboflavin responsive neuromuscular disorders by silencing flad-1 gene. Also, in the legend, we mentioned the cell types illustrated in the figure.
-Table 2: what is the tissue source (skin?) of the fibroblasts, iPSCs and MNs? Under Fibroblasts, is it “RTD patients carrying either SLC52A2 or SLC52A3 mutations”?
The tissue source of fibroblasts is indeed the skin, while iPSCs are derived from fibroblasts, by cell reprogramming. MNs are derived from iPSCs, following specific differentiation protocol. We added this information in revised Table 2. In Ref. 63, Authors studied two patients, carrying mutations in either SLC52A2 or SLC52A3 genes. We corrected the text accordingly.
-Line 444: which symptoms are improved or not by riboflavin administration?
It has been reported that riboflavin administration gradually improves symptoms, as muscle strength, motor function, respiration, hearing, and vision in some patients affected by RTD (Refs. 64, 65). We specified such effects in the revised version. However, the point is that not all the patients respond to therapy, nor the responders show amelioration of all symptoms. This high variability emphasizes the need to explore novel, custom therapeutic strategies.
-Line 480-482: summarize in 1-2 sentences the studies about antioxidants in RTD iPSCs and derived MNs.
We agree with the Reviewer, so we added a comment on these studies, in the revised version.
-Line 493: this section should be retitled “Perspective”. It should also be more concise and to the point.
We thank the Reviewer, and changed the title of the paragraph in the revised version of the manuscript. We have also improved this section by making it more concise.
-Finally, the authors did not address the question raised about including quantitative information when describing and comparing the mechanisms of disease implicated in Riboflavin Transporter Deficiency. At a minimum, the authors must indicate which effects are major effects or minor effects, preferably with supporting numbers and statistics, so that the readers can better understand the biological significance of these mechanisms.
We thank the Reviewer for his/her comment. For data quantitation, we refer to the cited experimental works by other Authors. As to major and minor effects, further studies are needed to clarify the correlation among pathomechanisms. However, to meet the Reviewer’s requirements, we tried to identify primary and secondary consequences of the genetic condition.